# Precision Imaging for Early Detection of Esophageal Cancer

**DOI:** 10.3390/bioengineering12010090

**Published:** 2025-01-20

**Authors:** Po-Chun Yang, Chien-Wei Huang, Riya Karmakar, Arvind Mukundan, Tsung-Hsien Chen, Chu-Kuang Chou, Kai-Yao Yang, Hsiang-Chen Wang

**Affiliations:** 1Division of Gastroenterology and Hepatology, Department of Internal Medicine, Ditmanson Medical Foundation Chia-Yi Christian Hospital, Chiayi 60002, Taiwan; 07742@cych.org.tw (P.-C.Y.); vacinu@gmail.com (C.-K.C.); 2Department of Gastroenterology, Kaohsiung Armed Forces General Hospital, 2, Zhongzheng 1st. Rd., Lingya District, Kaohsiung City 80284, Taiwan; forevershiningfy@yahoo.com.tw; 3Department of Nursing, Tajen University, 20, Weixin Rd., Yanpu Township, Pingtung County 90741, Taiwan; 4Department of Mechanical Engineering, National Chung Cheng University, 168, University Rd., Min Hsiung, Chiayi 62102, Taiwan; karmakarriya345@gmail.com (R.K.); d09420003@ccu.edu.tw (A.M.); 5Department of Internal Medicine, Ditmanson Medical Foundation Chia-Yi Christian Hospital, Chiayi 60002, Taiwan; cych13794@gmail.com; 6Obesity Center, Ditmanson Medical Foundation Chia-Yi Christian Hospital, Chiayi 60002, Taiwan; 7Department of Medical Quality, Ditmanson Medical Foundation Chia-Yi Christian Hospital, Chiayi 60002, Taiwan; 8Department of Medical Research, Dalin Tzu Chi Hospital, Buddhist Tzu Chi Medical Foundation, No. 2, Minsheng Road, Dalin, Chiayi 62247, Taiwan; 9Director of Technology Development, Hitspectra Intelligent Technology Co., Ltd., Kaohsiung City 80661, Taiwan

**Keywords:** esophageal cancer, hyperspectral imaging, object recognition, YOLOv5, squamous esophageal carcinoma

## Abstract

Early detection of early-stage esophageal cancer (ECA) is crucial for timely intervention and improved treatment outcomes. Hyperspectral imaging (HSI) and artificial intelligence (AI) technologies offer promising avenues for enhancing diagnostic accuracy in this context. This study utilized a dataset comprising 3984 white light images (WLIs) and 3666 narrow-band images (NBIs). We employed the Yolov5 model, a state-of-the-art object detection algorithm, to predict early ECA based on the provided images. The dataset was divided into two subsets: RGB-WLIs and NBIs, and four distinct models were trained using these datasets. The experimental results revealed that the prediction performance of the training model was notably enhanced when using HSI compared to general NBI training. The HSI training model demonstrated an 8% improvement in accuracy, along with a 5–8% enhancement in precision and recall measures. Notably, the model trained with WLIs exhibited the most significant improvement. Integration of HSI with AI technologies improves the prediction performance for early ECA detection. This study underscores the potential of deep learning identification models to aid in medical detection research. Integrating these models with endoscopic diagnostic systems in healthcare settings could offer faster and more accurate results, thereby improving overall detection performance.

## 1. Introduction

The diagnosis of esophageal cancer usually takes place in the latter stages when patients start to exhibit symptoms and seek medical treatment [1,2,3]. It is a highly fatal disease, accounting for more than 500,000 deaths each year [4]. In the year 2020, there were more than 0.6 million new cases and 0.54 million deaths worldwide [5]. By 2018, 283,433 people had died of esophageal cancer in China, accounting for 9% of the total incidence of cancer [6]. ECA’s geographical distribution distinguishes it from other types of cancer [7]. Males are more susceptible to it than females, and the prevalence rates fluctuate greatly [8,9], with a possible variation of around 100 times between the highest and lowest rates found in different nations [10,11,12]. Early detection of anomalies is crucial to promptly implement therapies and enhance the likelihood of successful therapy [13,14]. Nevertheless, the evaluation procedure might be difficult because of the subtle cellular alterations that take place throughout transitional stages and the subjective nature of their visual characteristics [15,16,17].

Hyperspectral imaging (HSI) is a widely used optical imaging technology in the field of medical image analysis and computer-aided diagnosis (CAD) [18]. HSI is a non-invasive [19], low-cost imaging method that uses a broad range of electromagnetic bands to evaluate optical tissue properties [20]. Tissue’s light interaction, namely photon scattering, is measured to generate spectrum images at precise spectral intervals, typically consisting of 100 or more images. Each image measures the proportionate absorption or reflection of light in a certain range of wavelengths, which can reveal biological characteristics like chromophores and tissue oxygenation [21,22]. This technology is distinctive as it integrates imaging with spectroscopy, allowing for the examination of a wide light spectrum range from visible to near-infrared wavelengths (380–1100 nm). HSI is superior to standard imaging due to its ability to collect spatial and spectral information in a 3D format, referred to as hypercube image data. Recent studies have proven that the integration of HSI with machine learning algorithms has been very effective in medical imaging [23].

Machine learning algorithms are effective instruments for discerning and categorizing cancer cells through the utilization of hyperspectral data [24]. M. Everson et al. utilized convolutional neural networks (CNNs) that were based on the intrapapillary capillary loops (IPCLs) classification of the Japanese Society of Endoscopy (JES). Their approach demonstrated a precision of 93.7% (with a range of 86.2% to 98.3%) in accurately differentiating abnormal IPCL patterns from normal ones. The accuracy of diagnosing aberrant IPCL patterns was 89.3% (with a range of 78.1% to 100%) for sensitivity and 98% (with a range of 92% to 99.7%) for specificity [25]. A study conducted by Masayasu Ohmori et al. directly evaluated the performance of a deep learning model with the diagnostic signs commonly employed by professional physicians. The model demonstrated a performance with an accuracy rate of 98%, a sensitivity rate of 68%, and a specificity rate of 83%. By contrast, physicians attained an accuracy rate of 82%, a sensitivity rate of 74%, and a specificity rate of 78% [26]. Hence, no substantial disparities were seen about gender, indicating that an artificial intelligence (AI) system could be employed as a valuable supplementary tool [27].

Narrow-band imaging (NBI) is a technique that uses small bands of blue and green light to selectively illuminate tissues, hence improving contrast, specifically targeting blood vessels and surface features within the mucosa. In contrast, HSI is capable of capturing images spanning the electromagnetic spectrum. This enables the acquisition of comprehensive spectral information, hence facilitating improved visualization and analysis. The utilization of both NBI and HSI techniques can effectively augment the imaging of blood vessels and lesions within the esophagus through a synergistic effect. While NBI relies on predefined spectral bands, HSI captures a broad spectrum, enabling the simulation of NBI-like effects through computational processing while retaining additional spectral data for enhanced diagnostic precision. The integration of different modalities enables a full portrayal and analysis of anomalies, which has the potential to enhance diagnostic precision and treatment strategizing in the diagnosis of esophageal cancer. Many studies have previously used HSI to detect various types of cancers. Courtenay et al. successfully employed robust statistical tests on HSI data in the VNIR spectrum to identify the hyperspectral differences between skin carcinomas and healthy skin [28]. Aboughaleb et al. proposed an HSI system to discriminate the tumor region from normal tissue of an ex vivo breast sample with sensitivity and specificity of 95% and 96% [29]. Urbanos et al. used support vector machine (SVM), random forest (RF), and convolutional neural network (CNN) machine learning algorithms to differentiate between healthy and tumor tissue during brain tumor surgery [30]. However, most HSI studies have used a hyperspectral imager or an NBI filter during the endoscope to generate hyperspectral images, which proves to be expensive.

Consequently, this work employs HSI-NBI and AI technologies to develop an AI-driven CAD system that can convert any WLI-RGB images into HSI-NBIs. Four different datasets based on WLI-RGB, WLI-HSI, NBI-RGB, and NBI-HSI have been trained on the YOLOv5 model to find out the best imaging modalities based on different parameters, including sensitivity, specificity, accuracy, F1-score, and κappa. This approach enhances diagnostic precision and assists doctors in identifying this potentially life-threatening illness. The main aim of this study is to improve the early detection of esophageal cancer by combining HSI with advanced AI models, thereby overcoming the significant limitations of current imaging techniques like NBI and WLI. Our objective is to address the deficiencies in visualizing early-stage lesions by utilizing the enhanced spectral resolution of hyperspectral imaging, along with sophisticated deep learning algorithms, to enhance diagnostic precision. This study presents an innovative methodology that integrates hyperspectral conversion technology with YOLOv5, a state-of-the-art object identification model, to establish an enhanced pipeline for the early diagnosis of esophageal cancer.

## 2. Materials and Methods

This study utilized hyperspectral conversion technology to enhance the training datasets, allowing the model to more effectively distinguish tiny differences in lesion shape and blood vessel patterns, in contrast to conventional methods. This study systematically compares the effectiveness of the RGB-WLI and HSI datasets in detecting early esophageal cancer, offering fresh insights into the efficacy of hyperspectral imaging in therapeutic applications.

### 2.1. Dataset

#### 2.1.1. Data Acquisition

The Ditmanson Medical Foundation Chia-Yi Christian Hospital and Kaohsiung Army General Hospital contributed a comprehensive dataset of white light endoscopy (WLI) and narrow-band endoscopy (NBI) images for this study. Images were collected from 255 patients, comprising 150 cases of esophageal cancer (ECA), 105 cases of dysplasia, and 80 cases of normal esophagus. Rigorous data quality checks were performed, discarding photographs with blurred focus, the presence of foreign objects, or duplicate captures to ensure optimal image quality. The final dataset included 3984 high-quality WLIs and 3666 NBIs, categorized into three groups: squamous cell carcinoma (SCC), dysplasia, and normal esophagus, based on pathological confirmation. To standardize the dataset, all images were resized to 608 × 608 pixels, ensuring consistency in the analysis.

#### 2.1.2. Image Processing

To enhance the diagnostic capability, WLIs and NBIs underwent a hyperspectral conversion process to extract 401 spectral bands covering the visible light range (380–780 nm). This step improved the visualization of vascular patterns and mucosal details, critical for detecting early-stage esophageal abnormalities. Preprocessing included data cleaning, image cropping to focus on regions of interest, and tagging with precise annotations by expert clinicians. Augmentation techniques such as rotation, flipping, and scaling were applied to diversify the dataset and prevent model overfitting. The amplification of under-represented categories ensured balanced class distributions.

#### 2.1.3. AI Processing

The YOLOv5 object detection model was employed to analyze the processed images. The dataset was partitioned into training (70%), validation (20%), and testing (10%) sets to assess the model’s performance. Hyperparameters, including learning rate and batch size, were optimized through iterative trials to achieve the best possible outcomes. The model was trained for 300 epochs, and its performance was evaluated using confusion matrices, precision, recall, F1-score, and accuracy metrics. These results demonstrated the model’s robustness and effectiveness in differentiating SCC, dysplasia, and normal tissues, even for subtle abnormalities.

### 2.2. Hyperspectral Imaging Algorithm

The primary objective of this study is to devise a snapshot-based VIS-HSI imaging algorithm capable of transforming an RGB image obtained by a point-and-shoot-based image acquisition device into a VIS spectral image. In order to achieve this goal, it is necessary to establish the correlation between the colors in the RGB image and a spectrometer. In order to establish this correlation, a Macbeth chart comprising 24 fundamental colors is employed as the target colors. The endoscope camera is configured to store the acquired image in the 8-bit JPG format, specifically in the sRGB color space ranging from 0 to 255. To facilitate the process of simplification, the values undergo a conversion into a reduced function ranging from 0 to 1, followed by a subsequent transformation into a linearized RGB value by the utilization of the gamma function. The conversion matrix (*M*) is utilized to convert the data into the CIE 1931 XYZ color space. On certain occasions, the camera may experience a range of defects, including nonlinear response, dark current, imprecise color separation of the filter, and color shift. For a comprehensive list of conversion formulas utilized in this study, please refer to Appendix A. To rectify these inaccuracies, a variable matrix (*V*) can be employed to derive the corrected values of X, Y, and Z (*XYZ_Correct_*), as seen in Equations (1) and (2). (1)$$\left[C\right]=\left[{XYZ}_{Spectrum}\right]\times pinv(\left[V\right])$$(2)$$\left[{XYZ}_{Correct}\right]=\left[C\right]\times [V]$$

The Macbeth chart is provided to an Ocean Optics (Dunedin, United States) QE65000 spectrometer in a controlled lighting situation to determine the reflectance spectra of the 24 colors. The determination of the brightness ratio (k) is derived from the standardized brightness value taken directly from the Y value of the XYZ color spectrum (XYZ_Spectrum_). Furthermore, the XYZ_Spectrum_ is transformed into the CIE 1931 XYZ color reference system. The procedure of constructing this algorithm is depicted in Figure 1.

The average root-mean-square error (RMSE) between *XYZ_Spectrum_* and *XYZ_Correct_* for the 24 colors is a mere 0.19, indicating an insignificant difference. Furthermore, the initial six principal components are derived through the application of principle component analysis (PCA) on the reflectance spectrum data (*R_Spectrum_*) acquired from the spectrometer, in conjunction with multiple regression analysis. In order to ascertain the correlation between *XYZ_Correct_* and *R_Spectrum_*, the dimensions of *R_Spectrum_* are diminished. Therefore, PCA is performed on *R_Spectrum_* in order to identify the primary principal components. Following the completion of the analysis, the initial six principle components were identified as the primary components capable of elucidating 99.64% of the observed variation in the data. The selection of the variable *V_Color_* in the regression analysis of *XYZ_Correct_* is based on its ability to encompass all potential combinations of XYZ values. Equation (3) is used to obtain the analog spectrum (*S_Spectrum_*) of the 24-color block, which is then compared with the *R_Spectrum_*. Prior to utilizing CIE DE2000 for the color difference calculation, it is necessary to convert *XYZ_Correct_* and *XYZ_Spectrum_* from the XYZ color space to the laboratory color space. The formula for conversion is as follows: (3)$$L^{*}=116f\left(\frac{Y}{Y_{n}}\right)-16\;\;\begin{array}{c} a^{*}=500\left[f(\frac{X}{X_{n}})-f(\frac{Y}{Y_{n}})\right]\\ b^{*}=200\left[f(\frac{Y}{Y_{n}})-f(\frac{Z}{Z_{n}})\right] \end{array}$$(4)$$f\left(n\right)=\left\{\begin{array}{c} n^{\frac{1}{3}},\;n>0.008856\;\\ 7.787n+0.137931,\;otherwise \end{array}\right.$$

The RMSE value for each of the 24 colors in the Macbeth chart is computed separately, resulting in an average deviation of 0.75. The observed result suggests that the recreated reflectance spectrum has a high level of similarity, with the colors being faithfully copied. (5)$${\left[SSpectrum\right]}_{300–700\;nm}=\left[EV\right]\left[M\right][V_{Color}]$$

### 2.3. YOLOv5

This study employed the YOLOv5 deep convolutional neural network model created by Ultralytics. This model utilizes a grid-based methodology, where the input image is partitioned into a grid of dimensions SxS. When an object’s true location is identified within a grid, that grid becomes responsible for detecting the object. The offset is determined by the disparity in the aspect ratio between the prior box (anchor) and the calculated value. The grid is categorized into either the backdrop or the item category by utilizing a predetermined threshold. Furthermore, the procedure entails including two neighboring grids that are situated at the center of the physical frame and align with the grid. In the end, the three grids together play a role in determining the detection result of the current frame.

The YOLOv5 model employs the displacement of the central location (t_x_, t_y_) and the proportional width and height (t_w_, t_h_) of the predicted frame with respect to the previous frame to produce predictions. The confidence score and categorization probability are computed concurrently. The ultimate forecast box probability is ascertained by conducting screening based on the IoU threshold and the maximum value of NMS.

The YOLOv5 neural network design comprises four primary components: input, feature extraction, feature analysis network, and prediction result output. The input stage utilizes three strategies to improve features: mosaic data augmentation, adaptive anchor box computing, and adaptive picture scaling. Mosaic data augmentation enhances the model’s capacity to accurately identify and locate minute objects inside images. The input data undergo a process of segmentation before being fed into the main network. The focus structure increases the number of channels in the initial three-channel images to 12. The CSP architecture minimizes parameters and model size by optimizing the design of the network structure. The neck module improves the fusion of network features by utilizing the FPN+PAN structure and incorporates larger feature maps to counterbalance the loss of information at the higher levels of the feature pyramid.

The YOLOv5 model employs the CIOU_Loss function to assess the recognition loss of the detected target rectangle during prediction. Appendix A provides in-depth explanations of the YOLOv5 detection algorithm and architecture.

## 3. Results

The YOLOv5 model was used to predict early esophageal adenocarcinoma (ECA) utilizing a total of 3984 white light images (WLIs) and 3666 narrow-band images (NBIs). The anticipated outcomes were categorized into three distinct groups: SCC, dysplasia, and normal. The training dataset was partitioned into four distinct subsets of data for model training: RGB-WLI, RGB-NBI, WLI hyperspectral pictures (HSI-WLI), and NBI hyperspectral images (HSI-NBI). Four distinct models were trained for every type of data.

The category’s ground truth box is shown in blue, while the predicted SCC category is represented by the red box, and the predicted dysplasia category is represented by the orange box. The prediction results for SCC are demonstrated using the white light ECA image detection model and the white light hyperspectral ECA image detection model in Figure 2a,b. The dysplasia outcome is predicted using the white light ECA image detection model and the white light hyperspectral ECA image detection model in Figure 2c,d. The findings suggest that the forecasted frame obtained from training with HSI-WLI exhibits more accuracy compared to training with WLIs.

Figure 3a,b depict the forecast of SCC outcomes utilizing the narrow-band ECA image detection model and the narrow-band hyperspectral ECA image detection model. The dysplasia outcomes are predicted using the narrow-band ECA image detection model and the HSI-NBI detection model in Figure 3c,d. The results indicate that training with HSIs produces prediction frames that closely mirror the real frames, surpassing the performance of conventional NBI training. This discovery offers proof that HSIs yield higher predictive outcomes.

The confusion matrix classification can be used to compute several metrics that analyze the deep learning approach’s capacity to reliably identify instances and evaluate its performance. The confusion matrix differentiates between true positives (TPs), false positives (FPs), false negatives (FNs), and true negatives (TNs) based on the current classification and the model’s predictions. The confusion matrix is visually depicted as a diagram, illustrating the association between the real and predicted occurrences.

The training results of the Yolov5 model are displayed in Table 1. The confusion matrix of the RGB-WLI model for white light ECA image recognition shows that out of 666 observed SCC frames, 521 were correctly predicted. Similarly, out of 744 dysplasia frames, 559 were accurately detected, and 642 out of the normal frames were correctly classified. There was a total of 514 correct predictions. Furthermore, 177 locations without definitive labels were accurately predicted, with 39 being classified as SCC, 75 as dysplasia, and 63 as normal. The overall accuracy rate is 78%, with a mean average precision (mAP) of 77%. The sensitivity for squamous cell carcinoma (SCC) is 78%, while the sensitivity for dysplasia is 75%.

The confusion matrix of the white light hyperspectral ECA image detection model shows that out of a total of 666 SCC frames, 589 were correctly predicted. Similarly, out of 744 dysplasia frames, 641 were accurately predicted, and out of 642 normal frames, 535 were correctly predicted. In addition, a total of 131 locations were identified without any distinguishing marks. Among them, 32 were classified as SCC (squamous cell carcinoma), 50 as dysplasia, and 49 as normal. The global accuracy rate stands at 86%, while the mAP is 83%. The sensitivity for squamous cell carcinoma (SCC) is 88%, and for dysplasia, it is 86%. The confusion matrix is essential for evaluating model performance, providing insights into true positives, false positives, and false negatives for all classes, including the background. Here, “background” refers to regions not belonging to SCC, dysplasia, or normal tissue. Background false positives (FPs) are non-class objects detected as one of the classes, while background false negatives (FNs) are missed objects classified as other background items. The matrix highlights the model’s strong differentiation capabilities across all categories. The amalgamation of HSIs with the YOLOv5 model shows a considerable enhancement in accuracy, precision, and recall relative to the NBI and WLI datasets, underscoring HSI’s capacity to surmount the constraints of conventional imaging methodologies. Our research tackles a significant issue in the identification of early-stage esophageal cancer—enhancing the visualization of tiny lesion features frequently overlooked in conventional imaging techniques.

Table 2 displays a comparative examination of evaluation markers for four distinct models. The markers, including AP, sensitivity, accuracy, and F1-score, are being compared between the SCC and dysplasia groups. The results indicate that the SCC category performs better than the dysplasia category in terms of these evaluation indicators. Dysplasia, as a premalignant condition, displays variations in color attributed to alterations in blood vessels, while SCC tumors infiltrate the epidermis, leading to irregularities in the tissue. Squamous cell carcinoma (SCC) exhibits more unique characteristics compared to dysplasia, as indicated by the higher values of AP (0.82, 0.88, 0.9, 0.91) and sensitivities (0.78, 0.89, 0.88, 0.92) in the two groups. The results indicate that the identification of SCC is superior to dysplasia.

The evaluation metrics mAP, sensitivity, F1-score, and total accuracy are utilized to evaluate the training results of WLIs and NBIs. The NBI data demonstrate superior performance compared to the WLIs in terms of mAP (0.87, 0.77), sensitivity (0.86, 0.78), F1-score (0.89, 0.83), and accuracy (0.86, 0.78). These results indicate that NBI training provides superior abilities in terms of detecting and distinguishing. The enhancement in performance can be related to the capability of the 415 and 540 nm bands, which are seen in NBIs, to emphasize vascular structures, thereby improving the overall model performance.

An analysis is conducted on the assessment indicators (mAP, sensitivity, F1-score, and accuracy) of HSI-WLIs and RGB-WLIs to compare the training outcomes of HSIs and RGB images. The findings suggest that the incorporation of HSI-WLIs results in a higher mAP of 0.83 compared to 0.77, greater sensitivity of 0.86 compared to 0.78, improved F1-score of 0.89 compared to 0.83, and enhanced overall accuracy of 0.86 compared to 0.78. Comparably, enhancements in mAP (0.87 to 0.9), sensitivity (0.86 to 0.9), F1-score (0.89 to 0.91), and overall accuracy (0.86 to 0.9) are noticed for HSI-NBIs and RGB-NBIs. These improvements are substantial and indicate that the transformation from white light photography to hyperspectral imagery produces superior outcomes compared to the transformation from narrowband imagery to hyperspectral imagery. Eliminating extraneous data and preserving crucial elements in WLI enhances overall performance enhancement. In contrast, the NBIs acquired at wavelengths of 415 and 540 nm emphasize the red signal produced by blue-green vascular fluorescence. While the conversion to HSI-NBIs helps reduce noise to some extent, it does not yield a significant enhancement in performance compared to white light conversion. Early diagnostics in this study refer to the detection of dysplasia, a precursor to esophageal cancer. The proposed HSI and AI-based method demonstrated significantly improved dysplasia detection compared to traditional approaches, highlighting its potential to enable timely intervention and improve treatment outcomes by identifying precancerous lesions with greater accuracy.

## 4. Discussion

Esophageal cancer (ECA) is one of the top 10 causes of cancer-related deaths in Taiwan. The identification of ECA presents a difficulty because there are no noticeable symptoms in the initial phases, and the mucosal changes are not easily noticeable [31]. The majority of malignancies, specifically over 85%, have their origins in epithelial tissue. Most epithelial malignancies are preceded by premalignant abnormalities that impact both the surface epithelium, which is the outer layer of cells, and the deeper stroma, which is the underlying tissue. These precancerous alterations disturb the ability of the tissue to absorb, scatter, and emit light, affecting its biochemical and structural characteristics [32,33,34]. Therefore, spectroscopy can be employed to detect these initial precancerous alterations. When light interacts with the tissue, it undergoes repeated episodes of elastic scattering, subsequent absorption, and fluorescence. Further scattering and absorption take place before the light ultimately leaves the surface of the tissue. Premalignant lesions lead to specific metabolic and structural modifications inside individual cells and their components, including changes in the ratio of nuclear material to cytoplasmic content and modifications in chromatin structure [35,36]. The application of spectroscopy provides immediate and accurate diagnostic information, resulting in significant cost savings associated with avoiding needless biopsies and treatment delays. Spectral data efficiently accentuate disparities between healthy tissue and elusive lesions, offering a novel approach for spotting ECA. This study addresses a significant gap in the field by illustrating the feasibility and efficacy of hyperspectral imaging for the early diagnosis of esophageal cancer, an area that has largely been neglected in prior studies. The noted enhancements in diagnostic performance highlight the capability of hyperspectral imaging to function as an adjunctive instrument in endoscopic diagnostic systems, tackling issues like inconsistent lesion presentation and unclear marking protocols. Our method offers a more detailed spectral profile of lesions, allowing clinicians to more effectively distinguish early malignant alterations, potentially resulting in enhanced treatment outcomes.

This study introduces a novel approach by developing a novel HSI technology which has the ability to convert a WLI into an HSI-NBI and combine it with YOLOv5 algorithms to enhance the accuracy of early-stage esophageal cancer detection, offering a comprehensive solution for improved diagnosis. The research findings reveal that HSI significantly improves the ability to differentiate between lesions, particularly when compared to WLI, with the highest level of performance observed with HSI-NBI. The validity of all four model assessments is confirmed by a κappa score exceeding 0.6. Additionally, the maximum mAP index and total accuracy rate reach a value of 0.9, suggesting potential for further improvement. As such, the model serves as a valuable supplementary tool to assist physicians in the early-stage diagnosis of ECA. Future efforts will focus on increasing the size of the datasets and enhancing the model’s prognostic efficacy to provide even greater support to physicians in the early detection of ECA. This study showcases a significant advancement in the timely identification of early-stage ECA through the integration of HSI and a diagnostic algorithm based on deep learning techniques. This approach significantly enhances diagnostic precision. This study demonstrates several strengths, including the integration of HSI and AI for the early detection of EC, which enhances prediction performance compared to traditional methods. The use of both white light and narrow-band images, along with HSI conversion technology, improves the visibility of blood vessels and lesions, contributing to more accurate diagnoses. Additionally, this study highlights the potential of deep learning identification models to aid in medical detection research, offering faster and more precise outcomes. However, despite the promising results, HSI also presents certain limitations and challenges. Firstly, HSI systems can be costly to acquire and maintain, potentially limiting their widespread adoption, especially in resource-constrained healthcare settings. Additionally, the complexity of analyzing hyperspectral data requires specialized knowledge and expertise, which may pose a barrier to implementation for healthcare professionals without a background in image analysis or machine learning. Furthermore, the large volume of data generated by HSI requires substantial storage and computational resources for processing and analysis, potentially leading to logistical challenges and increased processing times. Moreover, while HSI offers enhanced spectral resolution, its spatial resolution may be limited, potentially impacting the ability to detect small or subtle lesions. Additionally, the performance of HSI can be influenced by environmental factors such as lighting conditions and surface properties, which may introduce variability and affect the accuracy and reliability of the results. Despite these challenges, the integration of HSI and deep learning techniques holds great promise for improving the early detection and diagnosis of diseases such as ECA. Future research should focus on addressing these limitations through advancements in technology, algorithm development, and interdisciplinary collaboration, ultimately enabling the widespread and effective use of HSI in clinical practice for early disease detection and diagnosis. Future scope could also integrate spatial-spectral-based hyperspectral GAN (SSHGAN), which transforms hyperspectral images into standard histological images using networks trained by the cycle-consistent adversarial model [37].

This study highlights the significant advancements achieved through the integration of HSI and AI technologies for early-stage ECA detection. By employing the YOLOv5 model on datasets comprising WLIs and NBIs, the proposed method demonstrated notable improvements in accuracy, sensitivity, and precision. Specifically, the HSI-NBI approach outperformed traditional RGB-based methods, showing an 8% improvement in accuracy and a 5–8% enhancement in sensitivity and precision. These results underline the potential of HSI to enhance diagnostic accuracy by improving the visibility of blood vessels and lesions, offering clinicians a reliable tool for early detection. The proposed method stands out as it transforms WLIs into hyperspectral NBIs using hyperspectral conversion technology, which differs from existing methods that rely on spectrometers for data acquisition. This distinction limits direct comparisons with other hyperspectral methods in the literature, emphasizing the novelty and practicality of our approach. Unlike spectrometer-based systems, which are costly and complex, the proposed method is efficient and compatible with standard imaging modalities, making it more accessible for clinical implementation. This innovation has the potential to revolutionize early cancer detection by offering faster, cost-effective, and accurate diagnostic solutions for healthcare facilities.

## 5. Conclusions

In conclusion, this research study employed RGB white light and narrowband pictures, integrating hyperspectral conversion technology to enhance the visibility of blood vessels and lesions. As a result, the prediction performance of the training model was improved. An analysis of the RGB picture training model and the HSI training model showed notable enhancements in sensitivity, accuracy, precision, and recall. The HSI training model demonstrated an 8% improvement in accuracy, along with a 5–8% enhancement in precision and recall. As of now, there are no widely recognized criteria for labeling and processing images due to the variable color and shape of lesions, ambiguous instructions for marking frames, and significant discrepancies between doctors and patients. Moreover, there is currently no recognized treatment protocol for research connected to ECA. The esophagus’s distinctive environment poses specific obstacles, which justify the need for additional investigation. Gathering medical images is a difficult undertaking that requires the stringent safeguarding of patient confidentiality and an emphasis on acquiring accurate data for deep learning models. Enhancing the accessibility of data has the potential to optimize the efficiency of the model. Applying deep learning identification models to aid medical detection research shows potential for improving existing medical issues. These algorithms optimize efficiency for clinicians, allowing them to prioritize patient care. The timely identification of abnormalities enables prompt intervention and enhances treatment results. Integrating these models with endoscopic diagnostic systems at medical facilities can offer a plethora of information to improve detection capabilities, providing quicker and more precise outcomes for the diagnosis of early-stage esophageal cancer.

## Figures and Tables

**Figure 1 bioengineering-12-00090-f001:**
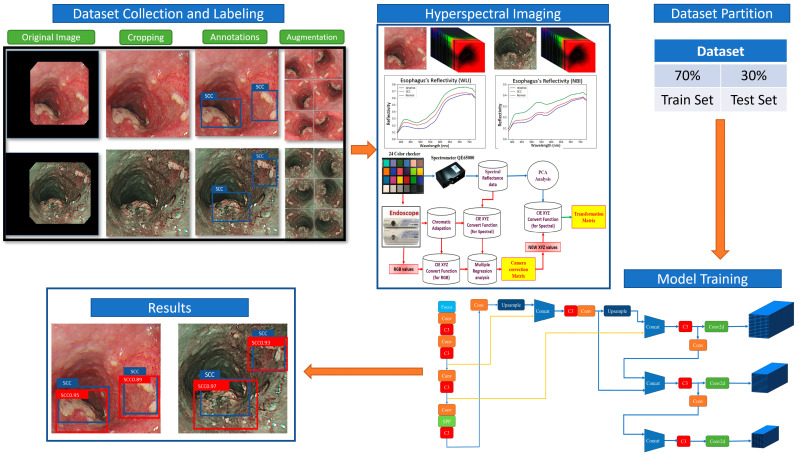
Experimental flow chart.

**Figure 2 bioengineering-12-00090-f002:**
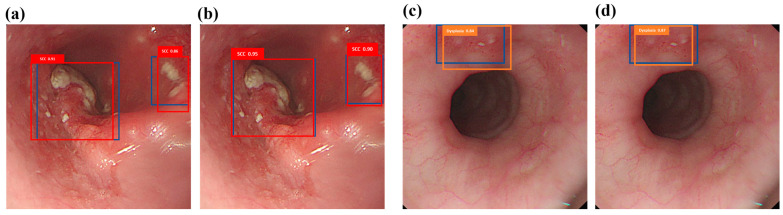
White light ECA image detection display: (**a**,**c**) are the results of white light ECA image detection models SCC and dysplasia, respectively, where the blue box is the real box position, and the red and orange boxes are the predicted boxes; (**b**,**d**) show the results of SCC and dysplasia categories of white light hyperspectral ECA image detection models. The blue box is the real box, and the red and orange boxes are the predicted boxes.

**Figure 3 bioengineering-12-00090-f003:**
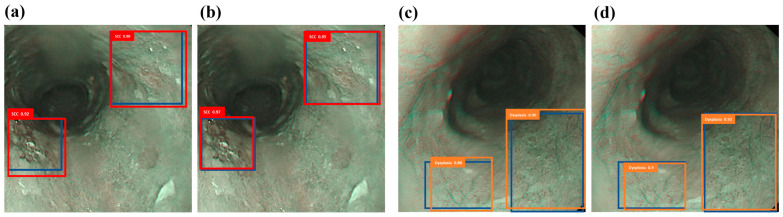
Narrow-band ECA image detection display: (**a**,**c**) are the results of narrow-band ECA image detection models SCC and dysplasia, respectively, where the blue box is the real box position, and the red and orange boxes are predicted boxes; (**b**,**d**) show the narrow-band hyperspectral ECA image detection model SCC and dysplasia category results. The blue box is the real box, and the red and orange boxes are the predicted boxes.

**Table 1 bioengineering-12-00090-t001:** Confusion matrix results of four models. mAp—mean average precision. RGB—red, green, and blue. WLI—white light imaging. NBI—narrow-band imaging. SCC—squamous cell carcinoma. HSI—hyperspectral imaging.

**RGB-WLI**						
	**True**	**SCC**	**Dysplasia**	**Normal**	**Background**	**Predicted All**
**Predicted**						
SCC		521	0	0	39	560
Dysplasia		0	559	0	75	634
Normal		0	0	514	63	577
Background		145	185	128	0	458
True all		666	744	642	177	2229
**HSI-WLI**						
	**True**	**SCC**	**Dysplasia**	**Normal**	**Background**	**Predicted all**
**Predicted**						
SCC		589	0	0	32	621
Dysplasia		0	641	0	50	691
Normal		0	0	535	49	584
Background		77	103	107	0	287
True all		666	744	642	131	2183
**RGB-NBI**						
	**True**	**SCC**	**Dysplasia**	**Normal**	**Background**	**Predicted all**
**Predicted**						
SCC		581	0	0	47	628
Dysplasia		0	571	0	42	613
Normal		0	0	526	71	597
Background		73	107	86	0	266
True all		654	678	612	160	2104
**HSI-NBI**						
	**True**	**SCC**	**Dysplasia**	**Normal**	**Background**	**Predicted all**
**Predicted**						
SCC		604	0	0	51	655
Dysplasia		0	594	0	55	649
Normal		0	0	546	63	609
Background		50	84	66	0	200
True all		654	678	612	169	2113

**Table 2 bioengineering-12-00090-t002:** Results of the four model evaluation indicators.

		Ap	Sensitivity	Precision	Specificity	F1-Score	Accuracy	κappa
RGB-WLI	SCC	0.82	0.78	0.93	0.96	0.85	0.78	0.62
	Dysplasia	0.71	0.75	0.88	0.93	0.81		
	Normal	0.79	0.8	0.89	0.94	0.84		
	Mean	0.77	0.78	0.9	0.94	0.83		
HSI-WLI	SCC	0.9	0.88	0.95	0.97	0.92	0.86	0.74
	Dysplasia	0.85	0.86	0.93	0.96	0.89		
	Normal	0.75	0.83	0.92	0.96	0.87		
	Mean	0.83	0.86	0.93	0.96	0.89		
RGB-NBI	SCC	0.88	0.89	0.93	0.96	0.91	0.86	0.72
	Dysplasia	0.85	0.84	0.93	0.96	0.88		
	Normal	0.89	0.86	0.88	0.94	0.87		
	Mean	0.87	0.86	0.91	0.95	0.89		
HSI-NBI	SCC	0.91	0.92	0.92	0.96	0.92	0.9	0.76
	Dysplasia	0.89	0.88	0.92	0.95	0.9		
	Normal	0.91	0.89	0.9	0.95	0.89		
	Mean	0.9	0.9	0.91	0.95	0.89		

## Data Availability

The data presented in this study are available in this article; further considerable requests can be made to the corresponding author (H.-C.W.).

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
