# Peer review of "Precision Imaging for Early Detection of Esophageal Cancer"

_bioengineering, 2025, doi:10.3390/bioengineering12010090_

Round 1
Reviewer 1 Report
Comments and Suggestions for Authors
Here the authors introduce a potential new role for YOLOv5 novel approach to assist clinicians in the detection of the early stages of Oesophageal Cancer. The use of this AI to differentiate between normal tissue, dysplasia SCC and ECA had reasonable success based on the samples provided and it will be expected that its accuracy will be greatly improved upon its future investment.
The comments are and suggestions are below;
Flow charts in the figures appear blurry, is the author able to provide clearer images?
How much more accurate is this AI technology (if so) compared to conventional methods? This should be discussed in the manuscript.
Reviewer 2 Report
Comments and Suggestions for Authors
This research is interesting for detecting Esophageal Cancer using HIS and AI technologies. However, there are essential remarks that need to be addressed.
1] Line 58: Citation 21 is irrelevant to this research.
2] Line 72: “IPCL” was not defined.
3] Line 77: 2 citations (30 and 31) to Ohmori et al. are the same.
4] Lines 105-108: NBI uses narrow spectral bands but HIS uses the whole spectrum. These are contradictory concepts.
5] Lines 118-121: there is again a description of AI methods after Lines 70-83. These descriptions should be merged.
6] Materials and Methods: there is no technical description of the basic instruments of the study, the endoscope, and the spectrometer. The basic hardware of the study should be described in more detail and added to the main text.
7] Line 133: ECA images were categorized as squamous ECA, dysplasia, and normal. But ECA images cannot be “normal” or “dysplasia”. Here, there is a fundamental misclassification. The authors should be careful with their categorical names. In addition, squamous esophageal carcinoma should be SECA or SEC, not SCC.
8] Materials and Methods: The basic steps of “Data acquisition”, “Image processing”, and “AI processing” are not clearly described and are not repeatable by other researchers.
9] Materials and Methods: It has already been published by some of the authors that NBI is superior to WLI (Kai-Yao Yang et al, Cancers, 2023). So, what is the point of using WLI again in this work?
Reviewer 3 Report
Comments and Suggestions for Authors
The early diagnosis of esophageal cancer represents a priority, as mortality and complications are severe in cases of late diagnosis. For this reason, I consider the article to be of interest, as it may contribute to improving the ease of diagnosis in the early stages.
The introduction would benefit from including specific data, such as the incidence of esophageal cancer, rather than general statements like, "Esophageal cancer (ECA) is one of the top 10 causes of cancer-related deaths in Taiwan." Additionally, it is currently too lengthy and overly focused on details that could be streamlined. Excessive bibliographic references in this section might overwhelm the reader. Consider relocating some of the detailed information to the Discussion section. Finally, the introduction should conclude with a clear and concise statement of the study's objective.
The sections on Materials and Methods and Results are well-written, clear, and appropriately detailed.
The Discussion section requires further expansion, particularly in terms of comparing the study’s findings with the existing literature. This would provide a broader context for the results and strengthen the overall interpretation and impact of the study.
Reviewer 4 Report
Comments and Suggestions for Authors
My comments and criticisms are in the attached file.

Round 2
Reviewer 4 Report
Comments and Suggestions for Authors
The authors did not address my main criticism, the applicability of machine learning for this kind of diagnostics. It was applied to a very specific set of high-quality pictures where the naked eye can make the conclusion. It is evident from all zeros at the confusion matrix's off-diagonal elements that the images of different outcomes are completely different. This will not help physicians in the diagnostics. A much more difficult and useful task will be obtaining the information from blurred images, which were excluded from the consideration. It would provide the conclusions readily obtained without waiting for histopathology results.
The paper in its current form should be rejected.
